# Deep Mining Generation of Lung Cancer Malignancy Models from Chest X-ray Images

**DOI:** 10.3390/s21196655

**Published:** 2021-10-07

**Authors:** Michael Horry, Subrata Chakraborty, Biswajeet Pradhan, Manoranjan Paul, Douglas Gomes, Anwaar Ul-Haq, Abdullah Alamri

**Affiliations:** 1Centre for Advanced Modelling and Geospatial Information Systems (CAMGIS), Faculty of Engineering and IT, University of Technology Sydney, Sydney, NSW 2007, Australia; Michael.J.Horry@student.uts.edu.au; 2IBM Australia Ltd., Sydney, NSW 2000, Australia; 3Earth Observation Centre, Institute of Climate Change, Universiti Kebangsaan Malaysia (UKM), Bangi 43600, Malaysia; 4Machine Vision and Digital Health (MaViDH), School of Computing and Mathematics, Charles Sturt University, Bathurst, NSW 2795, Australia; mpaul@csu.edu.au (M.P.); gomes@csu.edu.au (D.G.); anwaar.ulhaq@vu.edu.au (A.U.-H.); 5Department of Geology and Geophysics, College of Science, King Saud University, P.O. Box 2455, Riyadh 11451, Saudi Arabia; amsamri@ksu.edu.sa

**Keywords:** lung cancer, chest X-ray, malignancy predictive models, artificial intelligence, machine learning, computer vision, model mining

## Abstract

Lung cancer is the leading cause of cancer death and morbidity worldwide. Many studies have shown machine learning models to be effective in detecting lung nodules from chest X-ray images. However, these techniques have yet to be embraced by the medical community due to several practical, ethical, and regulatory constraints stemming from the “black-box” nature of deep learning models. Additionally, most lung nodules visible on chest X-rays are benign; therefore, the narrow task of computer vision-based lung nodule detection cannot be equated to automated lung cancer detection. Addressing both concerns, this study introduces a novel hybrid deep learning and decision tree-based computer vision model, which presents lung cancer malignancy predictions as interpretable decision trees. The deep learning component of this process is trained using a large publicly available dataset on pathological biomarkers associated with lung cancer. These models are then used to inference biomarker scores for chest X-ray images from two independent data sets, for which malignancy metadata is available. Next, multi-variate predictive models were mined by fitting shallow decision trees to the malignancy stratified datasets and interrogating a range of metrics to determine the best model. The best decision tree model achieved sensitivity and specificity of 86.7% and 80.0%, respectively, with a positive predictive value of 92.9%. Decision trees mined using this method may be considered as a starting point for refinement into clinically useful multi-variate lung cancer malignancy models for implementation as a workflow augmentation tool to improve the efficiency of human radiologists.

## 1. Introduction

### 1.1. Background

Lung cancer is the leading cause of cancer-related deaths worldwide [1], with over two million new cases documented in 2018 and a projected 2.89 million cases by 2030 [2]. There is a long history of research into the automated diagnosis of lung cancer from medical images using computer vision techniques encompassing linear and non-linear filtering [3], grey-level thresholding analysis [4], and, more recently, machine learning including deep learning techniques [5,6,7]. Despite many lab-based successes of computer vision medical image diagnostic algorithms, the actual regulatory approval and clinical adoption of these computer vision techniques in medical image analysis is very limited [8]. Clinical use of medical image AI is held back by the dissonance of evidence-based radiology [9] with the black-box nature of deep learning systems, along with data quality concerns and legal/ethical issues such as responsibility for errors [10]. As of September 2020, the U.S. Food and Drug Administration (FDA) has approved only 30 radiology related deep learning or machine learning based applications/devices, of which only three utilize the X-ray imaging mode [11], with the subject of one being wrist fracture diagnosis [12] and the other two being for pneumothorax assessment [13,14].

In contrast to the limited number of field applications relating to clinical use of machine learning in radiology, there exists a massive corpus of published research in this field [15,16]. The Scopus database [17] returns over 700 results for a title and abstract search on (“Computer Vision” OR “Machine Learning” OR “Deep Learning” AND chest AND X-ray). The overwhelming majority of these papers have been authored in the past decade, as shown in Figure 1.

Driving this huge interest in medical computer vision research is a desire to provide tools to improve the productivity of medical clinicians by providing an automated second reading to assist radiologists with their workloads [8]. This ambitious goal has technically been met by development studies under lab conditions [18], but persistent challenges including dataset size and diversity, along with bias detection/removal and expertise in oversight and safe use of such systems remain hinderances to clinical adoption [19].

### 1.2. Study Goals and Process Overview

The primary goal of this paper is to present a novel framework that automatically generates interpretable models for the stratification of lung cancer chest X-ray (CXR) images into benign and malignant samples. Rather than aiming to provide a narrow, automated CXR second reading using a feature extraction model for lung nodules as exhaustively discussed in [20], our objective is to autonomously create a range of reasonable and explainable decision tree models for lung cancer malignancy stratification using multiple pathological biomarkers for lung cancer as features. Our intention is that that these models can be used by the medical community as a data driven foundation for interpretable multivariate diagnostic scoring of lung cancer and form the basis of useful workflow augmentation tools for radiologists.

We extend the well-researched method of training a deep learning algorithm, typically a variant of the Convolutional Neural Network (CNN) architecture [21], in lung nodule detection and classification into a two-step approach that combines deep learning feature extraction with the fitting of shallow decision trees.

Firstly, we investigate lung pathology features that are considered biomarkers closely associated with lung cancer. We then use CXR examples of these pathologies to train a multi-class deep learning algorithm. Secondly, the score for each pathological feature is inferenced from the trained model for two independent lung cancer CXR datasets for which malignancy scoring metadata is available. Finally, this inferenced score tuple is fitted to the malignancy data for each patient using a shallow decision tree, with the most accurate decision trees extracted for discussion and further refinement. This process is illustrated in Figure 2.

This more holistic approach emphasizes the importance of multiple pathological biomarkers as lung cancer features, avoiding automation bias by the promotion of interpretability and expert human judgement in the creation of medical computer vision applications. It is hoped that this change of focus may help overcome the hurdles that have held back the adoption medical artificial intelligence algorithms into clinical workflow by enabling radiologists to oversee the advice provided by computer vision models, thereby promoting evidence-based and safe use of the technology [19].

### 1.3. Novelty and Major Contributions

Our key contributions stem from our novel combination of simple, proven techniques into an end-to-end process that mines interpretable models for the diagnosis and stratification of lung cancer. This process provides the medical community with two key novel elements. Firstly, we show that automated lung nodule detection from CXR alone is insufficient to indicate lung cancer malignancy. We therefore train our deep learning classifier on additional pathologies associated with lung cancer outside of the well-studied and narrow lung nodule classification task. This more closely matches the partially subjective workflow process of human radiologists [8] than other published studies. Secondly, our process provides an interpretable and logical explanation of model output for lung cancer malignancy stratification rather than the simple ‘black box’ score and visual, but not necessarily interpretable, saliency mapping that is common to other studies.

## 2. Related Work

### 2.1. Automated Lung Cancer Diagnosis

Although the Computed Tomography (CT) imaging mode has attracted most of research into machine learning based automated lung disease diagnosis, there are several studies that show the usefulness of CXR for this task. Best results against the very large ChestX-ray14 dataset [22] have been achieved using deep learning combined with trainable attention mechanisms for channels, pathological elements, and scale [23], achieving an average AUC of 0.826 for 14 thoracic conditions. Other recently published approaches include incorporation of label inter-dependencies via LSTM modules to improve prediction accuracy using statistical label correlations [24], augmenting deep learning with hand crafted shallow feature extraction [25] to improve classification accuracy over pure deep learning, and consideration of the relationship between pathology and location in the lung geometry as spatial knowledge to improve deep learning classification accuracy [26]. Each of these approaches improved upon the accuracy obtained from a pure deep learning approach by finding and exploiting additional information from the ChestX-ray14 dataset, however, none of these studies considered whether the additional information gleaned is generalizable. For instance, some label interdependencies may be specific to the clinic and radiologist population from which the ChestX-ray14 dataset was sourced and labelled. Likewise, the shallow feature extraction approach from [25] evaluated and tuned feature extraction algorithms to produce the best results against the ChestX-ray14 dataset, without experimenting to assess whether and to what extent this approach was generalizable to an independent dataset.

Very good results in lung nodule detection using deep learning have been achieved by teams using nodule-only datasets, with a systematic survey for this research being provided by [27]. State of the art lung nodule detection from CXR was achieved by X. Li et al. [6] using a hand-crafted CNN consisting of three dense blocks, each with three convolution layers. In this study, images from the Society of Radiological Technology (JSRT) dataset [28] were divided into patches of three different resolutions resulting in three trained CNNs, which were then fused. This scheme detected over 99% of lung nodules from the JSRT dataset with 0.2 false positives per image and achieved an AUC of 0.982. Despite these excellent results, this study did not stratify the detected nodules into malignant or benign categories, nor did this study perform generalization tests to confirm that the proposed model performed well against independent datasets.

Stratification of the JSRT dataset into malignant and benign nodule cases was achieved by [5] using a chained training approach. In this paper, a pretrained DenseNet-121 CNN was firstly trained on the ChestX-ray14 dataset [22] followed by retraining on the JSRT dataset. This model achieved accuracy, specificity, and sensitivity metrics for nodule malignancy of 0.744, 0.750 and 0.747, respectively. Once again, there were no generalization tests performed in this study.

Lung nodule classification studies typically utilize the Japanese Society of Radiological Technology (JSRT) dataset [28] as a machine learning training corpus. This dataset includes 150 samples with a nodule size of 3 cm or less, but only four samples with a nodule size greater than 3 cm. Use of the JSRT datasets in this manner is potentially problematic for two reasons; firstly, a lung nodule is defined as measuring ≤3 cm in diameter [29] with larger nodules or masses under-represented in these studies, even though these may indicate more serious and likely malignant cancers, and secondly, most pulmonary nodules are benign [30]. The CXR imaging mode is much more sensitive to calcified benign nodules (due to associated higher opacity) than non-calcified nodules or ground-glass opacities [31], which are more likely to be a sign of lung cancer [32]. These factors combined could lead to deep learning systems trained only on CXR nodule detection tending to under-diagnose serious nodules and/or masses over 3 cm in diameter, which is obviously undesirable. It is these concerns that have led us to investigate biomarker features other than visible nodules alone in the detection and malignancy stratification of lung cancer malignancy from CXR.

### 2.2. Automated Diagnostic Scoring

Interest in pathology severity scoring from CXR images has received much recent focus due to the COVID-19 pandemic commencing in 2020. A combination of deep learning feature extraction and logistic regression fit to severity has shown to be predictive of the likelihood of ICU admission for COVID-19 patients [33]. Many papers relating to classification and stratification of lung nodules detected from the CT imaging mode have been published employing various deep learning techniques [34,35], however few such papers have been published for the CXR imaging mode. This is due to the CXR imaging mode having lower sensitivity in comparison to the CT imaging mode [36], making nodule characterization and stratification difficult using segmentation and shape analysis techniques developed for higher resolution CT images.

The most comprehensive study into the use of CXR for pathological scoring was performed by [37], where deep learning was used to score long-term mortality risk from prostate, lung, colorectal and ovarian cancer across two randomized clinical trials. Patients were stratified into five risk categories (very low, low, moderate, high, and very high) based on a pre-trained inception [38]. CNN-based scoring of the patient’s initial CXR [37] concluded that the deep learning classifier was capable of accurate stratification of the risk of long-term mortality from an initial CXR. The study noted that most of these deaths were from causes other than lung cancer and speculated that the developed CNN and risk stratification reflected shared risk factors [39] apparent as biomarkers on CXR. This study shares our use of a CNN to score a range of pathologies for downstream generation of risk models, although it should be noted that our study aims to stratify lung cancer malignancy rather than long-term mortality.

## 3. Materials and Methods

### 3.1. Data Sourcing

To achieve our objective of automatically generating explainable lung cancer malignancy models, two logical datasets are needed. The first is a large corpus of labelled CXR data that can be used to train a deep learning classifier as a multi-pathology feature extraction component. The National Institute of Health ChestX-ray14 dataset [22] provides 112,120 frontal-view X-ray images of 30,805 unique patients. ChestX-ray14 images are uniformly 1024 × 1024 pixels in a portrait orientation with both Posterior-Anterior (PA) and Anterior-Posterior (AP) views. The second logical dataset must comprise CXRs with malignancy metadata, indicating whether lung cancer is present in the image and, if so, whether the cancer is considered by expert radiologists to be benign or malignant. There are two publicly available CXR datasets meeting these criteria. Firstly, the Lung Image Database Consortium Image Collection (LIDC-IDRI) [40] provides malignancy diagnosis metadata for 157 patient studies, of which 96 include PA view CXR images. Secondly, the Japanese Society of Radiological Technology (JSRT) [28] database provides 154 patient studies in the form of PA CXR images with malignancy diagnosis metadata. The JSRT dataset was labelled for lung nodule malignancy and subtlety by a panel of 20 experienced radiologists. The JSRT dataset is provided in Universal image format (no header, big-endian raw data), which was converted to Portable Network Graphics (PNG) format using the OpenCV [41] python library. The LIDC-IDRI dataset is provided in “Digital Imaging in Medicine” (DICOM) format and these files were converted to PNG format with the Pydicom [42] library using a grayscale colormap.

The LIDC-IDRI dataset has been manually labelled by four radiologists with access to corresponding patient CT scans. The label metadata has been provided at the patient level, meaning that there are some images provided where the nodule location is known and logged from the CT scan but not visible on the CXR image. Normally, any such inconsistency between the dataset and labels would be problematic for a computer vision diagnosis, since the image data would not support the label ground truth. Similarly, the JSRT dataset contains some nodules that are very subtle. For these subtle nodules of size 1–10 mm, human expert radiologist sensitivity was measured at only 60.4% [28]. Our proposed classification process should be relatively robust against these problems, since the presence of obvious nodules is only one of several lung cancer biomarkers under consideration.

### 3.2. Data Curation

The ChestX-ray14 data set was labelled using natural language processing to extract disease classes for each image from the associated radiology report, which the dataset authors report is of greater than 90% accuracy. Many of the images have a mix of disease classes. Since our objective is to achieve explainable lung cancer scores, we have restricted this study to images labelled with only a single disease class.

Of the 13 disease classes included in the ChestX-ray14 data set, not all are associated with lung cancer. To either exclude or include the classes, the simple rule was applied. If the literature noted a general indicative connection between lung cancer and the class in question, then that class was extracted from the ChestX-ray14 set for further analysis. The only exception to this inclusion rule is the “No Finding” class, which was included to enrich the generated models with a pathology contra-indicator. This resulted in five classes of interest for this study being Atelectasis, Effusion, Mass, No Finding, and Nodule. Once filtered in this way, the totals for images in this dataset are as shown in Table 1.

To address class imbalance during training, each class was under-sampled to 2000 examples of each class. The remaining class imbalance caused by the “Mass” and “Nodule” labels as minority classes (with 1367 and 1924 samples, respectively) was addressed in training by employing a weighted random sampler in the data loader.

Standard augmentations were applied only to the training ChestX-ray14 dataset with random rotation of 1 degree of expansion, and random horizontal flip. Vertical flipping was not used since CXR images were not vertically symmetrical. The images were resized with a default classifier size of 299 × 299 pixels for ResNet-50 [43] and ResNext-50 [44] and 244 × 244 pixels for other classifiers including DenseNet-121 [45], VGG-19 [46] and AlexNet [47]. Training and testing were run with and without equalization.

The ChestX-ray14 dataset was split into an 80:20 training and validation pair, resulting in 6641 images for training and 1661 images for validation. A set of 6085 images conforming to the official test split for ChestX-ray14 was used as a holdout test set. These images were drawn from the official test split to ensure that there was no patient overlap between the data used for training/validation and testing.

**Table 1 sensors-21-06655-t001:** Summary of ChestX-ray14 images extracted for deep learning.

Classification	Count	Extracted	Association
Atelectasis	2210	Y	Documented as a first sign of lung cancer [48].
Cardiomegaly	746	N	Not related to lung cancer although in rare cases misdiagnosed when underlying condition is mass in same geography of CXR [49].
Consolidation	346	N	Can sometimes accompany lung cancer but usually associated with pneumonia [50].
Edema	51	N	Can be a complication from treatment for lung cancer but does not indicate lung cancer [51].
Effusion	2086	Y	Can be caused by a build-up of cancer cells and a common complication of lung cancer [52].
Emphysema	525	N	Linked as a risk factor for lung cancer but not an indication [53].
Fibrosis	648	N	Linked as a risk factor for lung cancer but not an indication [54].
Hernia	98	N	Mistaken for lung cancer but does not indicate lung cancer [55].
Infiltration	5270	N	Generic descriptor used informally in radiological reports and not actually an accepted lung disease classification.
Mass	1367	Y	A primary indication of lung cancer [51].
No Finding	39,302	Y	Not lung (by definition) cancer but included to enrich generated models with a counter-indicator.
Nodule	1924	Y	A primary indication of lung cancer [51,56] with about 40% of nodules being cancerous.
Pleural Thickening	875	N	This is often an indication of mesothelioma caused by exposure to asbestos. It is also a very common abnormal finding on CXR. It is not an indication of lung cancer [57].
Pneumonia	176	N	Often a complication of lung cancer [50] with 50–70% of patients developing a lung infection. Persistent pneumonia can lead to a diagnosis of lung cancer. Not typically used as indicator of lung cancer.
Pneumothorax	1506	N	Can be the first sign of lung cancer but this is rare [58].

### 3.3. Model Development

#### 3.3.1. Network Selection

Following experimentation with a number of classifiers, including VGG-19 [46], AlexNet [47], DenseNet-121 [45], ResNet-50 [43] and ResNext-50 [44], we found that the DenseNet-121 and ResNet-50 networks initialized with ImageNet [59] weights consistently provided the equivalent and best results. We therefore selected the DenseNet121, and ResNet-50 network architectures for this study, which is consistent with other studies relating to the use of deep learning classifiers on large CXR datasets [23,60,61,62], with DenseNet being the most popular neural network architecture for lung CXR studies [63], and ResNet allowing for larger input images which would theoretically improve nodule localization. Noting that several state-of-the-art studies have employed network attention mechanisms in computer vision applications we additionally tested with a variant of ResNet-50 using a triple attention mechanism, which applies attention weights to channels and spatial dimensions using three separate branches covering channel/width, channel/height and width/height as described in detail by [64].

We followed standard practice employed in transfer learning [65] and replaced the network head fully connected layer (by default 1000 neurons) with the number of classification outputs required by the experiment being five. These five output nodes matched our five selected features being Atelectasis, Effusion, Mass, No Finding, and Nodule. This network was then fine-tuned using the training data subset of ChestX-ray14, achieving AUC-ROC results consistent with state-of-the-art for this dataset in consideration that we have restricted classes to PA view only and under-sampled (Table 2). All models converged well as shown in Figure 3, with the ResNet based networks being fully trained at 25 epochs compared to Densenet-121, requiring around 50 epochs to fully train.

Experimentation showed that the Adam optimizer [66] led to faster convergence and more accurate results (by around 5%) over the SGD [67] optimizer, therefore we chose to use the Adam optimizer with standard parameters (β1 = 0.9 and β2 = 0.999), along with a cosine annealing learning rate scheduler with an initial learning rate of 0.001 for Densenet-121 and 0.0004 for the Resnet-based classifiers. These rates were tuned by experimentation to produce the highest holdout test accuracy. The cosine annealing scheduler was selected because during model testing and hyperparameter optimization, it was noticed that the model trained well with a more aggressive learning rate, leading to higher validation accuracy at a lower number of epochs.

Upon inspection of the dataset images, we noticed a high degree of variation of brightness and contrast, both within and across the ChestX-ray14 and LIDC-IDRI datasets, as shown is Figure 4a,b. The JSRT dataset has consistent brightness and contrast since it was automatically equalized at extraction from raw image format, as evident from Figure 4c.

There was some concern regarding these differences in image brightness and contrast, which would confound deep learning model training. To minimize the inter and intra dataset differences in brightness and contrast, the models were trained and tested with standard histogram equalization as part of the image pre-processing pipeline. The experiments performed are summarized in Table 3.

#### 3.3.2. Deep Learning Model Performance

The results of 10 training/holdout testing runs are shown in Figure 5, Figure 6 and Figure 7. The holdout test split used was a subset of the recommended ChestX-ray14 test split containing the extracted classes, as listed in Table 1.

Excellent holdout testing results were achieved from all experiments. Test A achieved a maximum average AUC value of 0.789 at epoch 17. Tests B achieved a maximum average AUC value of 0.793 at epoch 5. Test C achieved a maximum average AUC value of 0.795 at epoch 10. Each of these results could be considered outliers from individual runs, and the average lines plotted in Figure 5, Figure 6 and Figure 7 give a better indication of real-world performance accounting for error margins, being 0.770, 0.771, and 0.777 for tests A, B and C respectively.

The overall best results were achieved by experiment C, with 4 models achieving average AUC above 0.790, and a superior mean profile in the range 10–20 epochs, which is interpreted as a reduced tendency to overfit, thereby promoting our objective of generating good decision tree fitted models in the downstream process. In contrast, experiment B resulted in only one model with average AUC above 0.790 and experiment A resulted in no single models with average AUC above 0.790.

Best AUC-ROC values for the extracted features for each tested configuration are shown in Table 2. The AUC-ROC values for the same conditions from the original ChestX-ray14 paper are also included as a baseline [22], along with the most relevant state-of-the-art results from [23], using a triple attention network with a DenseNet-121 backbone, and [24], which also considered additional pathologies in their multi-classifier, used mixed label images, and did not restrict the CXR images to PA projections. These studies also did not include the “No Finding” class in training or inferencing. Due to these differences in methods, our results are not directly comparable to these studies; the results have been compared only to establish that our deep learning models have comparable or better performance than similar, but not identical development models, and are a good basis for the following step of our process, being the fitting of shallow decision trees to these models.

#### 3.3.3. Deep Learning Model Attention via Saliency Mapping

The ChestX-ray14 dataset provides bounding box co-ordinate metadata for the pathologies present in a small subset of CXR images. These bounding boxes have been hand-labelled by a board-certified radiologist [22] and are useful for disease localization ground-truth comparison to classifier predictions. We used a Grad-CAM [68] visualization to compare our models’ predictions to ground truth from the ChestX-ray14 dataset bounding metadata with a selection of results shown in Figure 8a–p.

These results were generated from the model with the highest validation score from each experiment training round. Overall, the results showed a good correlation between the model’s predicted localization and the provided ground-truth. We noted that experiments B and C tended to produce the best localization results, with experiment C (being the ResNet-50 triplet attention network) providing the best localization performance overall, with particularly good results for the difficult “nodule” class, which was relatively poorly localized in experiments A and B using networks without attention mechanisms.

### 3.4. Malignancy Model Generation

Of the 157 patient studies from the LIDC-IDRI annotated with patient level diagnosis, 120 DICOM files contained both CT and CXR images, with the remaining 37 containing only CT scans. The 120 CXR images were extracted into a PNG format as earlier described to match the classifier input data format. Twenty-four of these images were labelled with a diagnosis of “Unknown” and were excluded from further analysis. The remaining 96 records were categorized by the LIDC-IDRI as follows in Table 4.

All 154 JSRT nodule CXR images were used in our experiments despite around 30% of the nodules in these images being classified by the JSRT as either “Very subtle” or “Extremely subtle”. Human radiologist sensitivity scores were relatively low for these images as 69.4% and 60.4%, respectively as reported by the JSRT source paper. The JSRT categorizes images as follows in Table 5.

For testing, the LIDC-IDRI and JSRT datasets were also combined by aligning diagnosis labels and assigning JSRT benign images to a diagnosis label of “1” and malignant images to a diagnosis label of “2”.

The models for each training epoch up to 25 epochs were used to extract pathological feature scores for the LIDC-IDRI, JSRT and combined image sets by inferencing. This resulted in a total of 250 fitted decision trees per experiment. A seven-column csv template was prepared containing columns for the Patient ID, placeholders for the five features of interest (including the “No Finding” class), and the diagnosis score 1 to 3 as determined by four experienced thoracic radiologists [69]. LIDC-IDRI diagnosis scores 2 and 3 were combined into a single malignancy class with 65 images representing malignant diagnosis and thereby allowing for a binary separation. Values for “Atelectasis”, “Effusion”, “Mass”, “No Finding” and “Nodule” were inferenced from the deep learning models as a score tuple and written to the placeholder columns to complete a data-frame of patients, inferenced feature scores and diagnosis labels.

The data-frame was then randomly split into an 80:20 training/testing set, before being used to fit a decision tree classifier with a limited maximum depth of 3 (to avoid overfitting due to the small sample size), fitting on an entropy criterion. The fit accuracy was captured and written to a CSV file, with the decision tree visualization captured as an image file for any model with greater than 60% accuracy for further investigation of the associated confusion matrix and tree as a potentially useful multivariate diagnostic and malignancy stratification model.

## 4. Results

The experiment generated many fitted decision trees meeting the stated accuracy objective of 60%. The process was especially effective for the combined LIDC-IDRI and JSRT datasets, which yielded 633 such trees from a total of 750 candidate trees, representing an 84.4% success rate. This is impressive considering that candidate trees were fitted for all epochs, including undertrained early epochs where poor fit results were concentrated.

Experiment C, based on ResNet-50 with triple attention network, proved to be the most consistent deep learning classifier, with 217 out of 250, or 86.8% success rate compared to 86.0%, and 80.4% for experiments A and B, respectively. This aligned well with our earlier deep learning holdout testing results presented in Figure 5, Figure 6, Figure 7 and Figure 8, where experiment C provided the best overall results. All fitted decision trees, along with confusion matrices for each tree, have been made available as supplementary material.

Recognizing that accuracy alone is of limited statistical value, especially for small and imbalanced datasets, we provide sensitivity, specificity, positive predictive value (PPV), false positive rate (FP Rate) and F1 value for each dataset as detailed in Table 6, Table 7 and Table 8 based on best fitted tree/s for each experiment as determined by interrogation of the confusion matrices for the fitted decision trees.

### 4.1. Model Analysis-Individual Datasets

The most accurate decision tree achieved 85.0% accuracy with sensitivity of 86.7%, specificity of 80.0%, positive predictive value of 92.9% with one false positive. This result was achieved for experiment from experiment A, using the LIDR-IDRI dataset. The confusion matrix for this result is shown in Figure 9a.

Inspection of this confusion matrix shows that this result, whilst promising, is based on a holdout test set of 20 images, being the 20% test split of the LIDC-IDRI dataset of size 96. Nevertheless, the decision tree fitting this result, as shown in Figure 10, seems reasonable.

Notably, a high score for the “No Finding” feature results in a branch that indicates a benign condition unless a high score for “Effusion” is present, whilst a lower score for the “No Finding” feature, along with a high score for the “Mass” feature results in a malignancy classification.

The JSRT based experiments resulted in lower decision tree fit metrics than the LIDC-IDRI tests. This is the expected result of our inclusion of subtle and very subtle nodule sample images. When reduced from a native image size of 2048 × 2048 pixels to the classifier default or 224 × 224 for DenseNet and 299 × 299 for the ResNet based classifiers, it is unlikely that these subtle nodules ranging from 1 mm to 15 mm diameter would still be visible as a detectable feature on the image. Nevertheless, we did achieve interesting results for this dataset with an overall best result from experiment C, which achieved 71% accuracy with good sensitivity of 100.0%, offset by limited specificity of 40%, positive predictive value of 64%, and a high false positive rate of 60%. The confusion matrix for this result is shown in Figure 9b.

Again, this result is based on a relatively small test set, being 31 holdout samples from the JSRT test set. The decision tree fitting for this result is shown in Figure 11.

The JSRT based model offers some interesting insights. Notably, the decision tree is rooted in “Effusion”, with high values for this feature associated with malignancy. Very low values for the “No Finding” feature in the presence of “Effusion” also lead to a malignancy classification. Once again, a high value for the “Mass” feature also leads to a malignancy classification.

### 4.2. Model Analysis-Combined Dataset

The best models from the individual LIDC-IDRI and JSRT datasets are promising but based on holdout test dataset of very limited size. To obtain the most statistically reliable results, we proceeded to fit decision trees against the combined LIDC-IDRI and JSRT datasets with aligned diagnosis labels.

The confusion matrices for the highest accuracy results of each experiment for the combined LIDC-IDRI and JSRT datasets is shown in Figure 12a–c, with associated metrics in Table 8. The best result for combined dataset testing was achieved by experiment C, using a DenseNet classifier enhanced with a triple attention mechanism. Experiment C represents the best balance between accuracy, sensitivity, and specificity with scores of 82%, 86% and 73%, respectively, resulting in a correct classification of 41 of the 50 test samples, with only 4 false positives (26.7%). Experiment B also performed relatively well, correctly classifying 38 of the 50 test samples with well-balanced accuracy, sensitivity, and specificity metrics of 76%, 80% and 67%, respectively, with 5 false positives (33.3%). Experiment A achieved a high accuracy of 82% by means of over-classification to the malignant class at the expense of specificity for the benign class, resulting in a high false positive rate of 60%.

Note that in the medical context, false positives are preferable to false negatives, since follow-up radiology, such as CT scans and biopsy analysis, will achieve a more accurate diagnosis [69,70,71] and eliminate the false positive. On the other hand, a false negative result can lead to a missed diagnosis and inaction, which is particularly problematic in the case of lung cancer, where early detection has been shown to significantly improve outcomes [72]. The best model from experiment C achieved an accuracy of 82%, with a sensitivity of 86% and specificity of range 73%, which (allowing for sensitivity/specificity trade-off) is consistent with studies showing human radiologist performance in detecting symptomatic lung cancer from CXR to have a sensitivity of 54–84% and specificity of up to 90% [70].

The proposed experimental method is novel, and there are no directly comparable studies against which we can compare these results. Studies presenting deep learning classifiers for the ChestX-Ray14, and other CXR datasets do not typically test the generalization characteristics of the presented models against independent datasets. Where studies such as [6] do assess cross-database performance, this is done by fine-tuning the model against the second datasets, thereby eliminating the independence of the second dataset and preventing use as a generalization study. We purposefully did not fine-tune our DenseNet-121 deep learning model against the LIDC-IDRI or JSRT datasets, preferring to preserve independence between the learning and inferencing datasets to promote unbiased cross-dataset feature extraction and realistic decision tree fitting.

### 4.3. Combined Dataset Decision Tree Interpretation

Example decision trees corresponding to the combined dataset results are shown in Figure 13a–c. These decision tree models explain the scores achieved by each test using a decision path and serve to illustrate the result of the end-to-end technique presented in this paper. Due to the small sample size of the combined LIDC-IDRI and JSRT datasets used for inference, it is not possible to claim that these decision tree models are clinically viable. However, even on this modest dataset, the results achieved are reasonable and could be expected to be greatly improved by additional inferencing samples. For example, all three decision trees are rooted at either the “Nodule” or “Mass” feature as expected, since these are the primary features for lung cancer.

The model for experiment A shown as Figure 13a indicates that a high score for “Nodule” with the presence of Effusion is associated with malignancy classifications. Conversely, a high score for the “No Finding” feature results in a benign classification. The decision tree generated for experiment B, in Figure 13b, shows a high score for “Mass”, which leads to a malignancy classification, whilst a high score for “No Finding” leads to a benign classification.

Experiment C yielded the best decision tree fitting results, as evident from Table 8. Therefore, we expected the decision tree for experiment C, as shown in Figure 13c, to provide the clearest insights into the relationship between our selected pathological features and lung cancer malignancy. This model shows that a high score for “Mass” combined with a low score for “No Finding” with “Effusion” leads to malignancy classification nodes. Low scores for “No Finding” with Nodule scores greater than 0.01 also lead to malignant classifications. A low score for “No Finding”, along with a high score for “Atelectasis” leads to a benign classification, reflecting the weak relationship between this feature and lung cancer, as found in our literature review. We intend to further refine these models using clinical studies and may filter out this feature in future iterations.

Interestingly, both experiments A and C associate “Effusion” with malignancy, with the decision tree for experiment A separating 73% of samples on this feature. This could indicate that the models based on the combined dataset were sensitive enough to automatically detect the build-up of fluid and cancer cells between the chest wall and lungs, associated with malignant lung cancer known as Malignant Pleural Effusion [73].

In general, we found the higher levels of the generated decision trees correspond to our understanding of the lung cancer condition, with the lower levels of the tree being less consistent and sometimes counter-intuitive, especially for decision tree nodes containing only a small number of samples.

## 5. Discussion

Using a novel hybrid of deep learning multiple feature extraction and decision tree fitting, we have automatically mined lung cancer diagnostic models that are capable of stratifying lung cancer patient CXR images from an independent dataset into benign/malignant categories. Our best model using a combined LIDC-IDRI and JSRT dataset achieved sensitivity and specificity of 85.7% and 73.3%, respectively, with a positive predictive value of 88.2%. Our best model using the LIDC-IDRI dataset alone achieved sensitivity and specificity of 86.7% and 80.0%, respectively, with a positive predictive value of 92.9%, using a smaller holdout test set which leads us to favour the lower but more statistically significant combined dataset result. These results are interpretable using human readable decision tree diagrams. The decision tree models provide explanations into lung cancer malignancy that may lead clinicians to consider factors that would otherwise be missed in a high-pressure environment, where, on average, radiologists may be required to interpret an image every 3–4 s in a workday [74].

We consider this multi-variate approach to be more useful than a narrow automated second reading for nodules only, since it aims to enhance the qualitative reasoning process undertaken by trained physicians, potentially increasing efficiency and reducing errors [8]. It is possible to conceive an implementation of our system as an augmented reality application presenting an interpreted overlay to radiologists, perhaps adapting to the clinician’s eye movements to ensure that attention is given to parts of the CXR that are important but have not yet captured the clinician’s attention.

The results presented in this research show the potential of hybrid machine learning computer vision techniques in automatically mining explainable, multivariate diagnostic scoring models from CXR image data. Whilst none of the developed models could be considered fit for clinical purposes at this stage, our malignancy classification results and good matching of decision tree structure to the medical literature, suggests that the technique we have developed is able to capture important pathological information and is worthy of further refinement and clinical trialling. Our technique provides an end-to-end process, resulting in clearly explainable insights that are amenable to expert oversight, thereby paving the way for clinical adoption.

## 6. Conclusions

Around 90% of missed lung cancer detections from radiological investigation stem from the CXR imaging mode [75]. Interpretable computer vision would provide a useful strategy to reduce radiologists’ observed error by broadening clinical attention to multiple abnormalities. Such tooling would also help to minimize missed diagnosis, resulting from early satisfaction of search [76], where obvious anomalies capture attention at the expense of more subtle abnormalities.

Given the small size of the LIDC-IDRI and JSRT datasets used for the experiments diagnostic ground truth, our results suggest that with additional data and further refinement of this method could be used to develop useful clinical methods to assist in the diagnosis and malignancy stratification of lung cancer.

Our future directions include utilizing state-of-the-art signal-to-noise improvement techniques applied to the CXR pre-processing pipeline, customization of the deep learning feature extraction algorithm to include wavelet filtering, followed by reference implementation in a federated deep learning framework to reduce patient selection bias and guarantee data privacy, thereby providing a path to clinical validation.

## Figures and Tables

**Figure 1 sensors-21-06655-f001:**
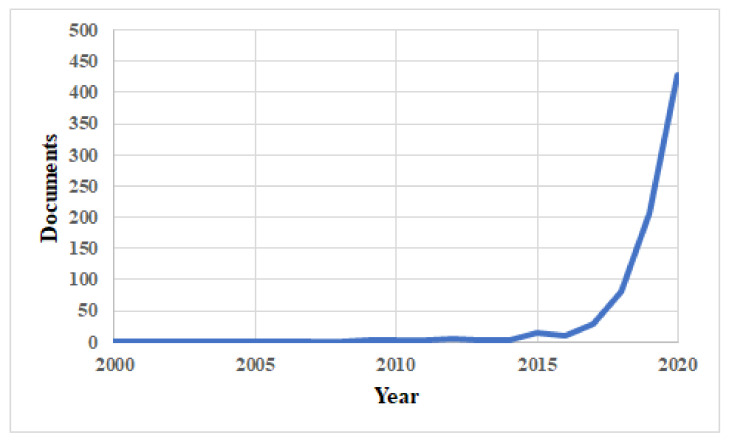
Scopus bibliographic histogram relating to machine learning/deep learning and chest X-ray showing exponential growth in published works over the past decade.

**Figure 2 sensors-21-06655-f002:**
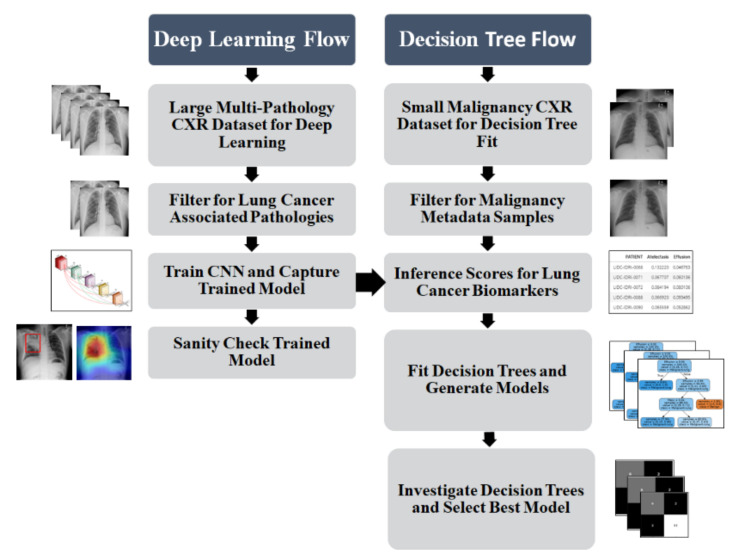
Automated lung cancer malignancy decision tree mining process showing deep learning flow for feature scoring, and decision tree flow for multi-variate model generation.

**Figure 3 sensors-21-06655-f003:**
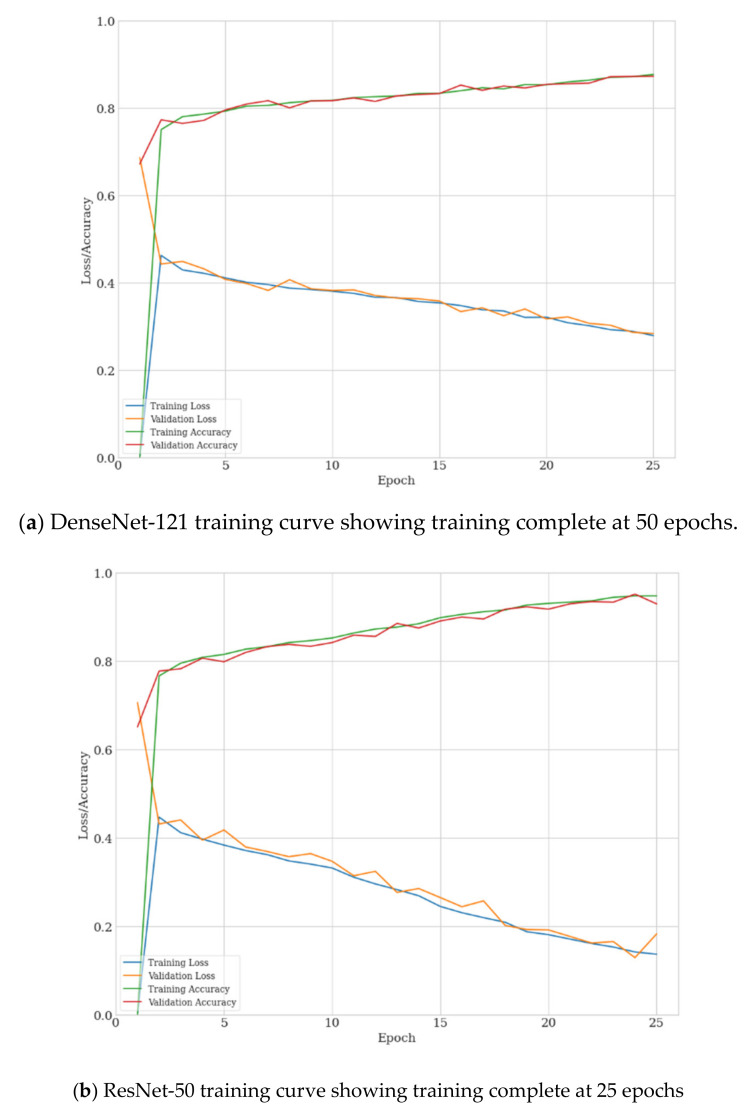
Training and validation loss/accuracy curves for DenseNet-121 (**a**), ResNet-50 (**b**), and ResNet-50 with Triplet Attention (**c**).

**Figure 4 sensors-21-06655-f004:**
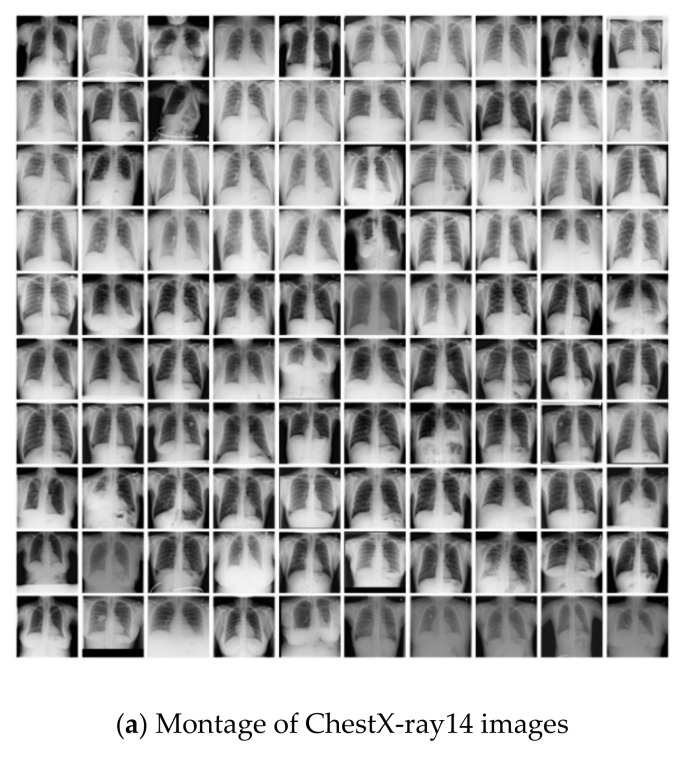
CXR montages for (**a**) ChestX-ray14, (**b**) LIDC-IDRI and (**c**) JSRT datasets showing inter and intra dataset brightness and contrast variation.

**Figure 5 sensors-21-06655-f005:**
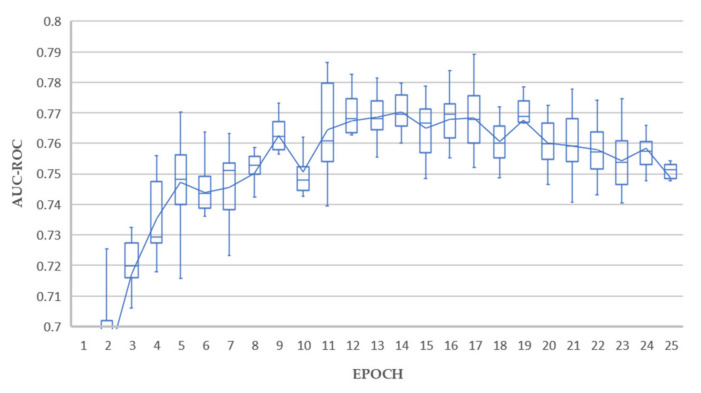
Average AUC-ROC for 10 rounds of holdout testing (Test A).

**Figure 6 sensors-21-06655-f006:**
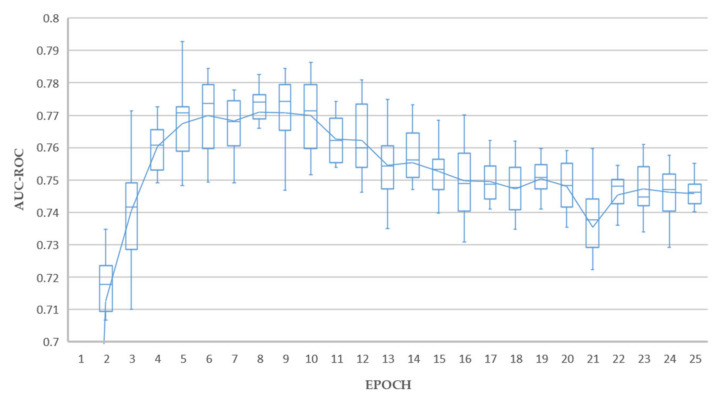
Average AUC-ROC for 10 rounds of holdout testing (Test B).

**Figure 7 sensors-21-06655-f007:**
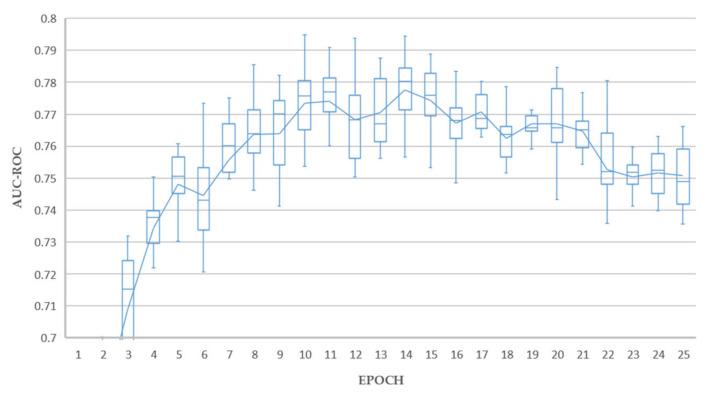
Average AUC-ROC for 10 rounds of holdout testing (Test C).

**Figure 8 sensors-21-06655-f008:**
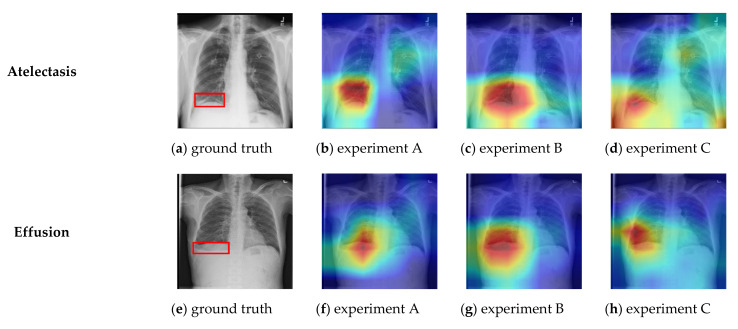
Ground truth comparison localization vs. Grad-CAM visualized predicted location for tested network architectures.

**Figure 9 sensors-21-06655-f009:**
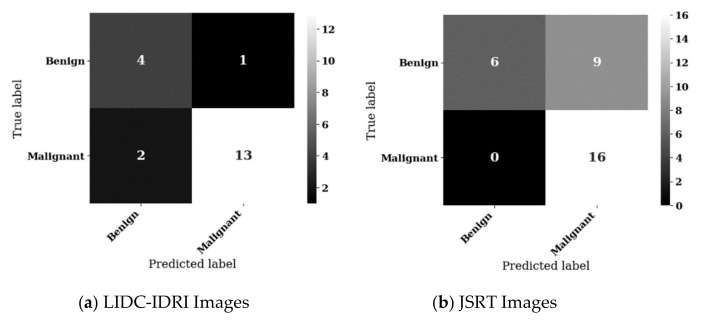
Confusion matrices for best results from LIDC-IDRI and JSRT inferencing datasets analysed individually.

**Figure 10 sensors-21-06655-f010:**
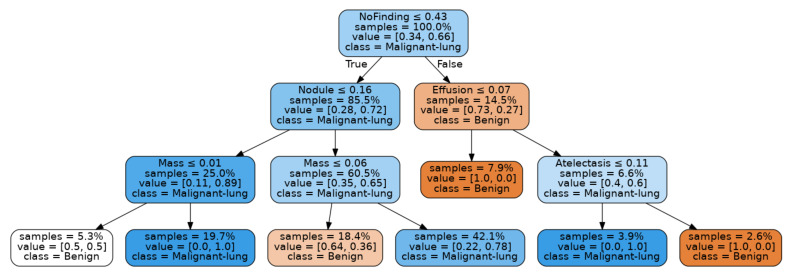
Decision tree fitting best result from LIDC-IDRI Images.

**Figure 11 sensors-21-06655-f011:**
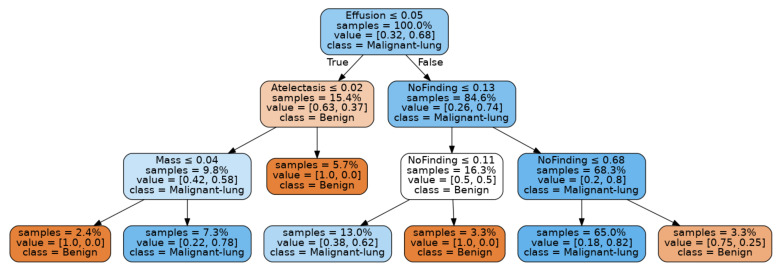
Decision tree fitting best result from JSRT Images.

**Figure 12 sensors-21-06655-f012:**
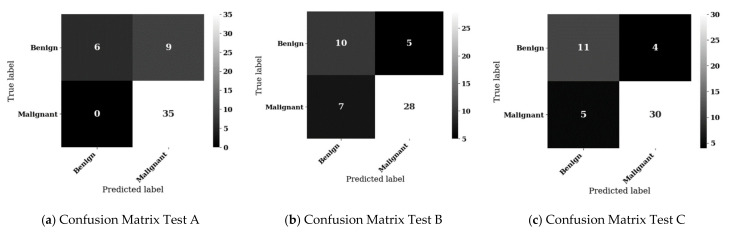
Confusion matrices for experiments A, B, and C using combined LIDC-IDRI and JSRT datasets.

**Figure 13 sensors-21-06655-f013:**
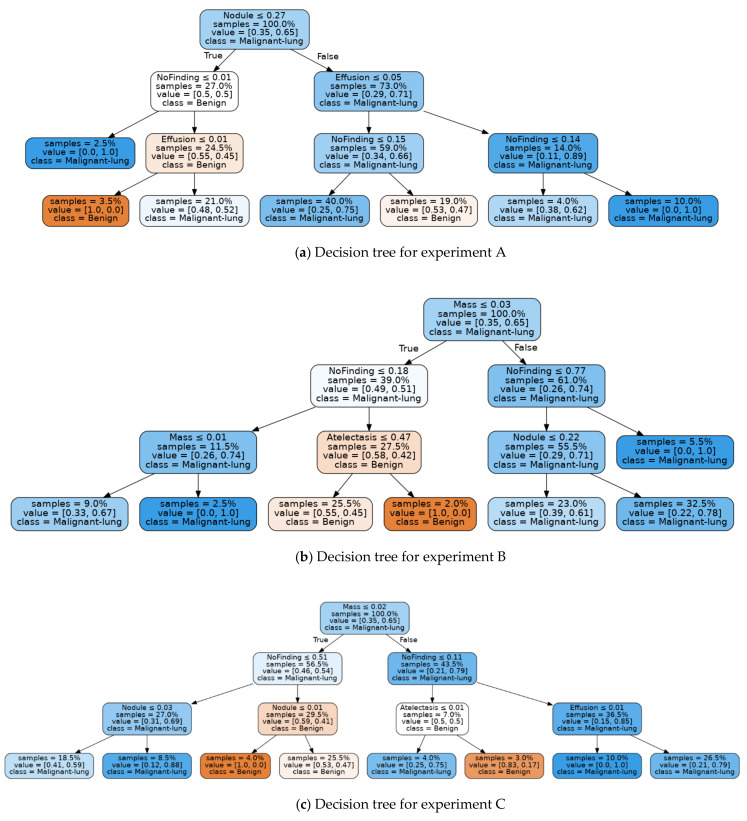
Best fit trees for combined LIDC-IDR and JSRT inferencing datasets for experiments A, B, C and D.

**Table 2 sensors-21-06655-t002:** Comparison of achieved AUC-ROC scores for ChestX-ray14 holdout test subset to other published works using similar techniques.

Configuration	Atelectasis	Effusion	Mass	Nodule
Test A (Epoch 17)	0.782	0.858	0.811	0.705
Test B (Epoch 7)	0.780	0.833	0.808	0.760
Test C (Epoch 8)	0.770	0.863	0.808	0.739
Wang et al. (2017) [22]	0.700	0.759	0.693	0.669
Wang et al. (2021) [23]	0.779	0.836	0.834	0.777
Yao et al. [24]	0.772	0.859	0.792	0.717

**Table 3 sensors-21-06655-t003:** Summary of Training/Test Scenarios.

Test	Network Architecture
A	Densenet-121 pretrained with ImageNet weights
B	Resnet-50 pretrained with ImageNet weights
C	Resnet-50 with Triplet Attention/pretrained with ImageNet weights

**Table 4 sensors-21-06655-t004:** LIDC-IDRI patient level diagnosis metadata summary.

Diagnosis	Description	Number of Images
1	Benign or non-malignant disease	31
2	Malignant, primary lung cancer	17
3	Malignant metastatic	48

**Table 5 sensors-21-06655-t005:** JSRT patient level diagnosis metadata summary.

Diagnosis	Description	Number of Images
Benign	Benign lung nodule	54
Malignant	Malignant lung nodule	100

**Table 6 sensors-21-06655-t006:** Summary of automatically generated decision tree metrics for LIDC-IDRI images.

Test ID	Accuracy (%)	Sensitivity (%)	Specificity (%)	PPV (%)	FP Rate (%)	F1
A	0.850	0.867	0.800	0.929	0.200	0.897
B	0.850	0.933	0.600	0.875	0.400	0.903
C	0.750	0.733	0.800	0.917	0.200	0.8148

**Table 7 sensors-21-06655-t007:** Summary of automatically generated decision tree metrics for JSRT images.

Test ID	Accuracy (%)	Sensitivity (%)	Specificity (%)	PPV (%)	FP Rate (%)	F1
A	0.677	0.938	0.400	0.625	0.600	0.750
B	0.677	0.875	0.467	0.636	0.533	0.737
C	0.710	1.000	0.400	0.640	0.600	0.781

**Table 8 sensors-21-06655-t008:** Summary of automatically generated decision tree metrics for combined LIDC-IDRI and JSRT images.

Test ID	Accuracy (%)	Sensitivity (%)	Specificity (%)	PPV (%)	FP Rate (%)	F1
A	0.820	0.100	0.400	0.796	0.600	0.886
B	0.760	0.800	0.667	0.849	0.333	0.824
C	0.820	0.857	0.733	0.882	0.267	0.870

## Data Availability

Publicly available datasets were analysed in this study. This data can be found at the cited locations.

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
