# Peer review of "Deep Mining Generation of Lung Cancer Malignancy Models from Chest X-ray Images"

_sensors, 2021, doi:10.3390/s21196655_

Round 1

Reviewer 1 Report

In this study, the authors approach an import topic, lung cancer detection. They use deep learning and decision trees to provide a classification of x-ray images. They worried about a very import issue, the explainability of the models. The manuscript is clear and well written, easy to follow, and with good figures, tables, and schemes. 

The background and related work is enough for understanding the study.

The datasets are well described. 

The results of the hybrid method seems to be better that others. 

Some issues:

  • Figure 3 side by side for better view;
  • Results section is number 3 and it should be number 4;
  • This study could be improved if the physicians were called for interpret the results.  

Author Response

Please see the file attached.

Reviewer 2 Report

Dear authors,

First of all, I would like to congratulate to the authors for this work. Absolutely, it is an interesting topic and very relevant using AI – Deep Mining Generation applied to Chest X-ray images. I was totally exhausted with the results proposing a model achieved sensitivity and specificity of 86,7% and 80% respectively with a positive predictive value of 92.9%. This paper is perfectly written, understandable, and well organized. I have learnt a lot with your proposal, and the literature review is excellent. The quality of the figures is excellent (for example, deep learning flow for feature scoring) that is clear the methodology used. Moreover, the description of the data curation is also relevant and it is useful for the results and discussion section. Finally, in this type of studies is important to use different strategies or methodologies and, in this work, demonstrates that the proposed solution has best results compared with the predecessors.

In general, my point of view is a that is a very high-quality work. For this reason, I accept this draft to be accepted for publish. Really, I hope that this work helps and improve this essential area in the health care fields.

Best regards,

Author Response

Please see the file attached.
